# Droplet Digital PCR for *Acinetobacter baumannii* Diagnosis in Bronchoalveolar Lavage Samples from Patients with Ventilator-Associated Pneumonia

**DOI:** 10.3390/antibiotics13090878

**Published:** 2024-09-13

**Authors:** Mirna Giselle Moreira, Anna Gabriella Guimarães Oliveira, Ihtisham Ul Haq, Tatiana Flávia Pinheiro de Oliveira, Wadi B. Alonazi, Antônio Augusto Fonseca Júnior, Vandack Alencar Nobre Junior, Simone Gonçalves dos Santos

**Affiliations:** 1Departament of Microbiology, Institute of Biological Sciences, Federal University of Minas Gerais, Av. Pres. Antônio Carlos 6627, Pampulha, Belo Horizonte 31270-901, Minas Gerais, Brazil; mirnamicro@ufmg.br (M.G.M.); annag@ufmg.br (A.G.G.O.); 2Department of Physical Chemistry and Technology of Polymers, Silesian University of Technology, M. Strzody 9, 44-100 Gliwice, Poland; ihaq@polsl.pl; 3Joint Doctoral School, Silesian University of Technology, M. Strzody 9, 44-100 Gliwice, Poland; 4Postgraduate Program in Technological Innovation, Federal University of Minas Gerais, Belo Horizonte 31270-901, Minas Gerais, Brazil; 5Federal Agricultural Defense Laboratory of Minas Gerais, Av. Rômulo Joviano s/n, Centro, Pedro Leopoldo 33600-000, Minas Gerais, Brazil; pinheirodeoliveiratf@gmail.com (T.F.P.d.O.); antonio.fonseca@agro.gov.br (A.A.F.J.); 6Health Administration Department, College of Business Administration, King Saud University, Riyadh 11421, Saudi Arabia; waalonazi@ksu.edu.sa; 7Interdisciplinary Center for Research in Intensive Care Medicine (NIIMI), Faculty of Medicine, Federal University of Minas Gerais, Av. Prof. Alfredo Balena 110, Santa Efigênia, Belo Horizonte 30130-100, Minas Gerais, Brazil; vandack@gmail.com

**Keywords:** *Acinetobacter baumannii*, droplet digital PCR, real-time PCR, diagnosis, ventilator-associated pneumonia

## Abstract

Advanced diagnostic technologies have made accurate and precise diagnosis of pathogens easy. Herein, we present a new diagnostic method, droplet digital PCR (ddPCR), to detect and quantify *Acinetobacter baumannii* in mini bronchoalveolar lavage (mini-BAL) samples. *A. baumannii causes* ventilator-associated pneumonia (VAP), a severe healthcare infection affecting patients’ lungs. VAP carries a high risk of morbidity and mortality, making its timely diagnosis crucial for prompt and effective management. Methodology. The assay performance was evaluated by comparing colonization data, quantitative culture results, and different generations of PCR (traditional PCR and Real-Time PCR—qPCR Taqman^®^ and SYBR^®^ Green). The ddPCR and qPCR Taqman^®^ prove to be more sensitive than other molecular techniques. Reasonable analytical specificity was obtained with ddPCR, qPCR TaqMan^®^, and conventional PCR. However, qPCR SYBR^®^ Green technology presented a low specificity, making the results questionable in clinical samples. DdPCR detected/quantified *A. baumanni* in more clinical samples than other methods (38.64% of the total samples). This emerging ddPCR technology offers promising advantages such as detection by more patients and direct quantification of pathogens without calibration curves.

## 1. Introduction

Nosocomial infections represent a major clinical challenge for diagnostic microbiology and significantly threaten public health safety. *Acinetobacter baumannii* is one of the highly reported pathogens associated with nosocomial infections, including ventilator-associated pneumonia (VAP), having an adaptive capacity to acquire and maintain multiple genetic elements encoding antimicrobial resistance determinants [1,2]. VAP mainly affects mechanically ventilated patients in intensive care units (ICUs). VAP develops at least 48 h after an endotracheal intubation or tracheostomy and is caused by pathogens not present or incubating during mechanical ventilation. VAP is characterized by progressive infiltrate signs of systemic infection, including elevated body temperature, altered blood cell count, and changes in sputum characteristics [3]. Recent reports of VAP infection show that the infection incidence ranges from 15% to 50% [4,5]. Thus, accurate and rapid identification of *A. baumannii* is critical for appropriate infection control in ICUs [6].

The conventional polymerase chain reaction (PCR) is commonly used for the detection of *A. baumannii*; however, it has some limitations, such as the requirement of an electrophoretic run on an agarose gel, absence of quantitative determination, and the use of harmful reagents such as ethidium bromide [7]. Real-time PCR (qPCR), also known as 2nd generation PCR, was developed in 1992 as an enhancement of conventional PCR and represents a significant advance in biotechnology for diagnosing infectious diseases [8]. However, its precision can be affected by amplification efficiency variations and inhibitor presence. Thus, the journey of PCR reached droplet digital PCR (ddPCR), one of the latest PCR techniques. ddPCR showed good efficiency in the detection of *A. baumannii* and *K. pneumoniae* in blood samples within 4 h, with a specificity of 100% [9]. Additionally, ddPCR has also proven to be efficient for diagnosing and quantifying *Pneumocystis jerovecii* in bronchoalveolar lavage fluid samples [10]. Clinically important pathogens, including *Mycobacterium tuberculosis* [11] and *Staphylococcus aureus* [12], are also successfully detected by ddPCR. The other potential applications of ddPCR are the identification of antimicrobial resistance genes in bacteria, confirmation of detection of SARS-CoV-2, expression levels of genes associated with the immunological synapse, and measurement of miRNA levels related to inflammation and aging [13].

ddPCR enables the precise detection and absolute quantification of nucleic acids, eliminating the requirement for a calibration curve [14]. The key to ddPCR’s enhanced performance lies in its ability to partition a sample into thousands to millions of separate reactions on microfluidic chips or within micro-droplets. These partitions are made at such a scale that each contains either a single or no copy of the target DNA or RNA molecule [9]. After PCR amplification, analyzing the proportion of partitions that yield a positive signal against the total number of partitions allows for directly calculating the absolute target nucleic acid quantity in the original sample. This calculation uses binomial Poisson statistics, enabling precise measurement of target copy numbers [15]. ddPCR showed promising results in the detection of *A. baumannii* [9]; therefore, we attempted to extend its diagnostic breadth towards the detection of VAP-associated *A. baumannii*, which is a life-threatening infection the timely diagnosis of which may reduce the mortality and morbidity rate.

This study aimed to develop a new ddPCR to detect and quantify *A. baumannii* in bronchoalveolar lavage samples from patients with VAP. In addition, we evaluated the performance of this assay by comparing quantitative culture results with those of different generations of PCR (traditional PCR, SYBR^®^, and Taqman^®^).

## 2. Results

### 2.1. Microbiological Analysis

A total of 44 microorganisms were isolated from mini-BAL samples. *A. baumannii* was the most prevalent, isolated from eight clinical specimens (18.2%) in quantitative culture. The second most recovered microorganism of the cases evaluated in this study was *S. aureus* (15.9%), followed by *Pseudomonas aeruginosa* (13.6%). All isolates of *A. baumannii* were resistant to all antimicrobials tested, except Polymyxin B. The mortality rate among the included patients was 20 (45.5%).

### 2.2. Analytical Sensitivity

The limit of detection for conventional PCR was 1.02 ng/µL (reference sample) or 2.00 ng/µL (clinical sample) (dilution 10^3^). For SYBR^®^Green, the detection limit was 0.010 ng/µL or 20 pg (reference sample) or 0.020 ng/µL or 40 pg (clinical sample) (dilution 10^5^). The limit of detection for TaqMan^®^ qPCR and ddPCR was 0.0010 ng/µL or 2 pg (reference sample) or 0.0020 ng/µL or 4 pg (clinical sample) (dilution 10^6^). Only the sigmoid curves were considered positive. We considered cutting off the 37 quantification cycle (Cq) for SYBR^®^Green and 35 for Taqman^®^. Table 1 shows the Cq of the reference *A. baumannii* ATCC 19606 and clinical sample dilutions close to the detection limit of both real-time PCR, values of absolute quantification by ddPCR, and its respective coefficient of variation (CV). Figure 1 presents the analytical sensitivity of ddPCR represented by serial dilution of *A. baumannii* ATCC 19606 type strain.

### 2.3. Analytical Specificity

Conventional PCR demonstrated 100% specificity against the panel of bacteria tested; only *A. baumannii* was amplified, and no non-specific amplification was visualized against the tested target blaOXA-51. With respect to the SYBR^®^ Green methodology, we verified the non-specific amplification of six bacteria (*S. aureus* ATCC 25923, *Klebsiella pneumoniae* ATCC 27799, *Escherichia coli* ATCC 25922, *Staphylococcus epidermidis* ATCC 29213, *Pseudomonas aeruginosa* ATCC 27853, and *Klebsiella oxytoca* 700324), through the existence o melting temperatures close to that of the positive control (*A. baumannii* ATCC 19606) used (Figure 2).

There was reasonable specificity of TaqMan^®^ against the target tested; non-specific amplification was not detected, except for *S. aureus* ATCC 25923 with C.T. values of 22.77 and 22.27 (duplicates). The same results were found in the ddPCR specificity; no significant counting drops were observed about the positive control (*A. baumannii* ATCC 19606). *S. aureus* ATCC 25923 showed some droplets in ddPCR (fluorescence amplitude below 6000); however, the amplitude was lower than the positive control. In this sense, we used the fluorescence amplitude 6000 as a cut-off to distinguish between positive and negative droplets (Figure 3).

### 2.4. Comparison of Conventional PCR, Real-Time PCR (SYBR^®^ Green and TaqMan^®^), and ddPCR on Detection of A. baumannii in Clinical Samples

Conventional PCR detected *A. baumannii* in 8 of 44 samples (18.18%) with a correlation of 69.44% (0.95 Confidence Interval, error 0.1183) between the traditional technique of PCR and quantitative culture). The SYBR^®^Green technique did not discriminate between negative and positive quantitative culture results and amplified all 44 samples with Cq 10.08 up to 34.19. Taqman^®^ amplified 11 of 44 (25%) (correlation of 66.67% between Taqman PCR assay and quantitative culture with 0.95 confidence interval, error 0.1125). This assay showed Cq values higher than 35 in 24 samples; however, the amplification curves were not sigmoid. ddPCR showed 17 positive samples (38.64% of the total samples) with a correlation of 41.55% about the golden standard (0.95 confidence interval, error 0.1096).

Of the eight samples detected in the quantitative culture, Taqman^®^ and ddPCR detected seven, and the conventional PCR detected six. One sample was detected in a quantitative culture but not in any molecular method (sample 5). Table 2 shows the molecular assays and quantitative culture evaluation results in clinical samples. Table 3 presents the quantity of detected and undetected samples by conventional PCR, Taqman, and ddPCR compared with quantitative culture.

In terms of a comparison of the ddPCR results versus conventional PCR and Taqman^®^ in clinical samples, the ddPCR showed more positive results, and the correlation was better with Taqman^®^ (69.23%, error 0.0921, 0.95 confidence interval) than conventional PCR (52.17%, error 0.1036 with 0.95 confidence interval) (Table 4).

## 3. Discussion

The criteria for diagnosing VAP are controversial, with various methodologies described in the literature for epidemiological surveillance and diagnosis [16,17]. Therefore, it is crucial to have consistent criteria and local data to accurately identify and manage VAP and its etiology.

Lower respiratory tract specimens (e.g., BAL, mini-BAL) are preferred for microbiological diagnosis of VAP, and detecting microorganisms in this sample can result in an effective and more accurate antimicrobial therapy [18]. In our study, the pathogen most frequently isolated in mini-BAL samples from 44 patients with VAP was *A. baumannii*, followed by *S. aureus* and *P. aeruginosa*. Similar results are reported in many studies that identify *A. baumannii* as a primary pathogen in VAP patients [19,20,21]. All isolates of *A. baumannii* exhibited resistance to the tested antimicrobials, except for Polymyxin B, indicating a high level of drug resistance consistent with previous reports [2,22]. This high drug resistance is associated with the presence of the carbapenemase gene *bla*OXA-51, which is responsible for the multi-drug resistance capacity of *A. baumannii* [23,24].

It is important to highlight that to evaluate colistin susceptibility, we chose the VITEK II system, even though broth microdilution is considered the gold standard as recommended by the joint CLSI-EUCAST Polymyxin Breakpoints Working Group. This decision was driven by the need for an efficient and rapid system capable of handling a large volume of samples in a practical and reliable manner, as our study was conducted primarily in a hospital routine laboratory. Recent studies have shown that VITEK II exhibits excellent categorical agreement with broth microdilution for most tested pathogens, including *A. baumannii* [25].

Currently, detection methods of *A. baumannii* include phenotypic characterization, which previously needed isolation, growth in culture medium, and incubation time [26]. In this context, rapid methods that can make the diagnosis quickly appear as alternatives in the scenario of ICUs. Molecular approaches to detect bacterial DNA in VAP have been used successfully, and recent studies use qPCR to diagnose VAP etiologic agents [27,28,29]. However, most studies explore other etiologic agents like *S. aureus* [28,30] and *S. pneumoniae* [31]. Moreover, ddPCR has yet to be studied and used for this purpose. This is the first study that used ddPCR to detect bacterial DNA in BAL from patients with VAP.

The analytical sensitivity of ddPCR showed a higher limit of detection (LOD) when compared with the two other PCR techniques (Conventional PCR and S.Y.B.R. qPCR). However, the analytical sensitivity of ddPCR was similar to the TaqMan qPCR; this similarity between ddPCR and TaqMan qPCR was also reported by Pinheiro de Oliveira et al. in the detection of the foot-and-mouth disease virus [32]. Our results indicated that ddPCR and TaqMan^®^ possess high analytical sensitivity, capable of detecting very small amounts of DNA in mini-BAL and reference samples, with concentrations as low as 1 pg/µL and 2 pg/µL, respectively. ddPCR showed reasonable specificity for all the tested bacteria except *S. aureus* ATCC 25923, which demonstrated fluorescence amplitude below 6000. This result was essential to better determine the cut-off of the ddPCR.

We detected a low analytical specificity for SYBR^®^ Green; this is a DNA intercalating agent, and the detection is monitored by fluorescence intensity over the cycles. It was suggested that this may be a significant drawback of the method since the dye is not specifically dye-bound to the double strand of target DNA [33], which may explain the non-specificity of the results. In the original work on these primers, Gadsby et al. [34] also used a probe in which a specificity step detected a positive signal only for *A. baumanni* DNA and did not detect amplification in any of the tested isolates found in the respiratory tract or commensals. The TaqMan system uses probes, which increase the sensitivity and specificity of the reaction; however, to obtain a quantitative result, the standard curve and the use of a reference gene for quantitative analysis by qPCR are essential to obtain reliable results [33,35,36,37].

Among the 44 mini-BAL samples from patients with VAP, ddPCR amplified 17 samples (38.64%), making it a more effective method for detecting positive samples compared to other molecular methods (conventional PCR and TaqMan) and culture techniques. ddPCR detected seven out of eight samples identified by culture, yielding a concordance rate of 41.55% with the quantitative culture results, because it detected bacteria in more samples. Notably, sample 5, which was identified by culture, was not detected by any of the PCR methods.

These results highlight the potential of ddPCR to replace traditional bacterial culture methods, providing a faster and equally reliable alternative for diagnosing VAP. ddPCR can rapidly and reliably confirm the involvement of *A. baumannii* in VAP, significantly reducing the diagnostic time by eliminating the need for bacterial culture growth. This expedited process allows for earlier intervention and more targeted treatment, potentially improving patient outcomes and reducing the duration of antibiotic therapy.

## 4. Materials and Methods

### 4.1. Criteria for Diagnosis of VAP

Ventilator-associated pneumonia (VAP) is diagnosed in patients with a pulmonary infection who have been on mechanical ventilation for at least 48 h. The diagnosis is confirmed when there is a new infiltrate or progression of a previously existing infiltrate on chest X-ray, in association with at least one of the following criteria:A mini-BAL culture showing a count of ≥10^4^ CFU/mL;An axillary temperature of ≥38.3 °C;Total leukocytes of ≥10^9^/L;Aspiration of purulent sputum through the orotracheal tube (OT) or tracheostomy; worsening of gas exchange, such as a PaO_2_/FiO_2_ ratio of less than 300;Increased need for supplemental oxygen or heightened ventilatory demand.

### 4.2. Sample Collection and Microbiological Analysis

Between January 2023 and December 2023, samples of mini-BAL were collected from 44 patients (23 men and 21 women, mean age 61 ± 19 years) with pneumonia associated with mechanical ventilation (VAP), admitted to an Intensive Care Center at Hospital das Clínicas/UFMG. The main comorbidities presented by these patients were arterial hypertension (22.72%), neoplasias (18.2%), and renal and hepatic insufficiency (11.4%). Before the sampling, the project was approved by the Teaching, Research, and Extension Directorate—DEPE—of Hospital das Clinicas and the Ethics and Research Committee.

Each mini-BAL sample obtained was homogenized and divided into five aliquots of approximately 2 ml using Eppendorf tubes. Direct microscopic analysis was performed using a smear, stained with Gram staining technique. The same aliquot was used for primary sowing in Tryptone Soya Agar (TSA-OXOID^®^) culture media for isolation of all pathogens and MacConkey (OXOID^®^) for isolation of Gram-negative rods. Using a 1 µL calibrated loop and after homogenization, the samples were cultured and incubated at 37 °C. The plates were evaluated for bacterial growth for 18–24 h; if there was growth, the number of colonies observed per mL on each plate was counted. The count was adjusted accordingly, given that the most widely accepted cut-off value in the literature is 10^4^ CFU/mL.

<10 colonies per plate <10^4^ CFU/mL and, presumably, colonization.≥10 colonies per plate ≥10^4^ CFU/mL and should be interpreted as infectious disease.

Gram staining was performed for all colonies in order to confirm the purity of the isolated bacteria. To identify the isolated bacteria(s) grown, the VITEK^®^ II system (BioMérieux^®^, Marcy-l'Étoil, France) was used employing cards (GN for Gram-negative and GP for Gram-positive) according to the manufacturer’s instructions. After identification by VITEK^®^ II, three batches of inocula were preserved in cryotubes containing 1 mL of Brucella Broth (OXOID^®^, Basingstoke, UK) plus 10% sterile Glycerol and stored in a −80 °C freezer.

Antimicrobial susceptibility testing was performed by an automated microdilution VITEK^®^ II (BioMérieux^®^) according to the recommendations of the Clinical and Laboratory Standards Institute [38]. Antimicrobials tested for *A. baumannii* were gentamicin (CN), ceftazidime (CAZ), Cefepime (FEP), ceftriaxone (CRO), imipenem (IPM), meropenem (MEM), Ampicillin/sulbactam (SAM), Piperacillin/Tazobactam (TZP), Ciprofloxacin (CIP), Colistin (COL), Tigecycline (TIG), Ampicillin (AMP), Cefuroxime (CRX), Cefuroxime Axetil (CRX), and Cefoxitin (CFO). Susceptibility to Polymyxin B (PB) was tested using the Etest ^®^ (AB Biodisk, Solna, Sweden).

### 4.3. Nucleic Acid Extraction

DNA from the 44 mini-BAL was extracted using the QIAamp^®^ DNA. Mini Kit (Qiagen, Hilden, Germany) according to the manufacturer’s guidelines. The extracted DNA was kept in a freezer at −80 °C. The presence and integrity of the extracted DNA were assessed by electrophoresis on an agarose gel. The reference bacterial DNA was extracted according to Moreira [39]. Each sample was tested in triplicate to evaluate the analytical specificity of the molecular assay. All the extracted DNA was quantified in Nanodrop (Thermo, Waltham, MA, USA) using the purity ratios.

### 4.4. Conventional PCR Assay

The DNA extracted was amplified by standard PCR using a Mastercycler Nexus Gradient thermal cycler (Eppendorf^®^, Hamburg, Germany) and OXA-51-like β-lactamase (blaOXA-51-like) species-specific primers (Forward 5′-TAA TGC TTT GAT CGG CTT TG-3′ and Reverse 5′-TGG ATT GCA CTT CAT CTT GG-3′ [40]) that generate a fragment with 353 bp. The 25 μL reaction consisted of 12.5 μL 2× Master Mix (PROMEGA, Madison, WI, USA), 1 μL of primer forward (10 μM), 1 μL of primer reverse (10 μM), 2 μL of DNA (50 ng/μL), and 8.5 μL of ABD-type water. The PCR conditions were performed as follows: 1 cycle of 94 °C for 3 min; 35 cycles of 94 °C for 45 s, 60 °C for 40 s, and 72 °C for 50 s; and one cycle of 72 °C for 5 min. The amplified products were electrophoresed on a 1% agarose gel, stained with gelRed (Sigma, St. Louis, MO, USA), and visualized under UV light.

### 4.5. SYBR^®^ Green qPCR Assay

qPCR was performed using Rotorgene Q (Qiagen) to amplify the blaOXA-51-like gene. Primers were designed to generate a 134 bp fragment, with the sequences Forward 5′-TTTAGCTCGTCGTATTGGACT-3′ and Reverse 5′-CCTCTTGCTGAGGAGTAATTTT-3′, as described by [34]. High-purity DNA was extracted using a Qiagen DNA extraction kit, and the concentration was determined using a NanoDrop spectrophotometer. The reaction mixture, with a total volume of 20 μL, included 12.5 μL of 2× SYBR^®^ Green PCR Master Mix, 1 μL each of forward and reverse primers (both at 10 μM concentration), 1 μL of DNA template at a concentration of 20 ng/μL, and 4.5 μL of nuclease-free distilled water. Optimal annealing temperature was determined through gradient PCR. The standardized PCR protocol started with a 10 min denaturation step at 95 °C, followed by 40 cycles of 15 s at 95 °C for denaturation and 60 s at 60 °C for annealing and extension. Controls included no template controls (NTC) to check for contamination, positive controls to verify reaction functionality, and a standard curve generated using serial dilutions of a known concentration of the target gene to determine efficiency and quantify unknown samples. Each reaction was run in triplicate for accuracy. Data acquisition was performed using qPCR analysis software. The threshold was set in the exponential phase of amplification, and baseline correction was applied to remove background fluorescence. Efficiency was calculated from the standard curve, aiming for an efficiency range of 90–110% (slope of −3.1 to −3.6).

After the 40 cycles, a melting curve analysis was conducted by gradually increasing the temperature from 60 °C to 95 °C, with fluorescence measurements taken every 0.3 °C increment to assess amplification specificity. A single peak in the melting curve confirmed the specificity of the qPCR amplification for the blaOXA-51-like gene.

### 4.6. TaqMan^®^ Real-Time PCR Assay (qPCR)

The oligonucleotides for TaqMan^®^ qPCR targeting the blaOXA-51-like gene (89 bp) were designed using the Primer3 Plus program [41]. The amplification was performed using the CFX96 Touch Real-time PCR Detection System (Bio-Rad Laboratories, Hercules, CA, USA). The reaction mixture, with a total volume of 20 μL, consisted of 10 μL of QuantiTect^®^ Probe Master Mix (Qiagen), 1 μL of probe (10 pmol) with the sequence 5′/56-FAM-CCTTGAGCA/ZEN/CCATAAGGCAACCAC/3IABKFQ-3′, 0.75 μL of forward primer (10 pmol) with the sequence 5′ CCTGCTTCGACCTTCAAAATG-3′, 0.75 μL of reverse primer (10 pmol) with the sequence 5′-TGCCCGTCCCACTTAAATAC-3′, 1 μL of MgCl2 (25 mM), 4.5 μL of nuclease-free water, and 2 μL of DNA template. The thermal cycling conditions were optimized as follows: initial denaturation at 95 °C for 10 min, followed by 45 cycles of 95 °C for 30 s and 60 °C for 60 s.

### 4.7. Droplet Digital PCR (ddPCR) Assay

The same oligonucleotides used in TaqMan^®^ Real-time PCR were used in ddPCR. The genetic material extracted was quantified using the QX200™ Droplet Digital™ PCR System (Bio-Rad Laboratories) while following the manufacturers’ instructions. Briefly, the reaction was carried out using 10 μL ddPCR Supermix for probes, 0.6 μL probe (10 pmol), 2.16 μL of each forward and reverse primer (10 pmol), 7.08 μL of H_2_O and the 2 μL of DNA. After, 20 μL of the PCR reaction was placed into an 8-channel disposable droplet generation cartridge and 70 μL of droplet generation oil was added per well. Approximately 20,000 droplets were generated at each well by the QX200 droplet generator). After processing, the prepared droplet emulsion (40 μL) was further transferred to a skirted 96-well PCR plate (Eppendorf). The plate was heat-sealed with pierceable foil using the PX1™ PCR plate sealer (Bio-Rad). PCR amplification was carried out using a Px2 Thermal Cycler (Thermo Electron Corporation, Waltham, MA, USA). The reactions were first optimized in terms of the primer-annealing step (in the temperature range between 54 and 60 °C). The amplification conditions were as follows: 95 °C for 10 min; 40 cycles of 94 °C for 30 s and 58 °C for 60 s; and 1 cycle of 95 °C for 10 min. After amplification, the PCR plate was read by the QX200 droplet reader (Bio-Rad Laboratories) and analyzed by Quanta Soft droplet reader software, version 1.6.6.0320.

### 4.8. Analytical Sensitivity

To compare the sensitivity and accuracy among the three PCR generations, the DNA of 10 × 10^4^ CFU/mL of *A. baumannii* ATCC 19606 type strain (1099 ng/μL) was extracted according to Moreira [13] and diluted by a factor of around 10 (101 to 1015). The detection limit was also defined using serial dilution of a clinical sample (10 × 10^4^ CFU/mL–2002 ng/μL). Dilutions close to the limit were repeated in triplicate (real-time) or duplicate (ddPCR), and we considered the analytical sensitivity as the last dilution amplified or quantified in all replicates by the methods.

### 4.9. Analytical Specificity

To verify the analytical specificity of the three generations of PCR, a panel of eight reference bacteria that are commonly isolated species in quantitative culture in bronchoalveolar lavage samples from patients with ventilator-associated pneumonia was used: *S. aureus* ATCC 25923, *K. pneumoniae* ATCC 27799, *Klebsiella oxytoca* ATCC 700324, *Enterococcus faecalis* ATCC 29212, *E. coli* ATCC 25922, *Enterobacter cloacae* ATCC 700323, *S. epidermidis* ATCC 29213, and *P. aeruginosa* ATCC 27853. The four molecular methods were used in 44 mini-BAL DNA samples. The amplification or quantification (ddPCR) results were also compared with the results for *A. baumannii* in quantitative culture as the “golden standard”, in addition to the colonization data of the patients with *A. baumannii* at the time of collection.

### 4.10. Statistical Analysis

All statistical analyses were performed with the software package SPSS. version 19.0 (IBM, Armonk, NY, USA). Linear regression analysis of the calibration curves from real-time PCR was conducted using two different software platforms: Rotorgene Q Series Software version 2.1.0 for SYBR. green assays and Bio-Rad CFX Manager version 3.1 for Taqman assays. The values of the quantification cycle (Cq) obtained in analytical sensitivity were evaluated in triplicate (SYBR. and Taqman) or duplicate (ddPCR) for each test considering the coefficient of variation (CV). For ddPCR, the absolute concentration of each sample was automatically reported by QuantaSoft version 1.6.6.0320 software by calculating the ratio of the positive droplets over the total droplets combined with Poisson distribution. The error reported for a single well was the Poisson 95% confidence interval. The kappa statistic with quadratic weighting [42], available at http://vassarstats.net/kappa.html (accessed on 15 March 2024), was used to establish the correlation between the ddPCR and other molecular assay results with quantitative culture in clinical samples.

## 5. Conclusions

DdPCR and qPCR Taqman^®^ showed better analytical sensitivity toward VAP associated with *A. baumanni*. However, ddPCR was more efficient than the other techniques in clinical samples, so it can be considered a promising molecular tool for VAP diagnosis to detect and estimate the bacterial load, mainly when absolute quantification is required. This work made a new tool available for the diagnosis of *A. baumanni*. Herein, we present ddPCR as an important diagnostic method for the detection of VAP-associated *A. baumannii*, having several exceptional features such as precision, high sensitivity and specificity, absolute quantification and quantification in a complex mixture, less vulnerability to inhibitors, and no requirement for post-amplification processing.

## Figures and Tables

**Figure 1 antibiotics-13-00878-f001:**
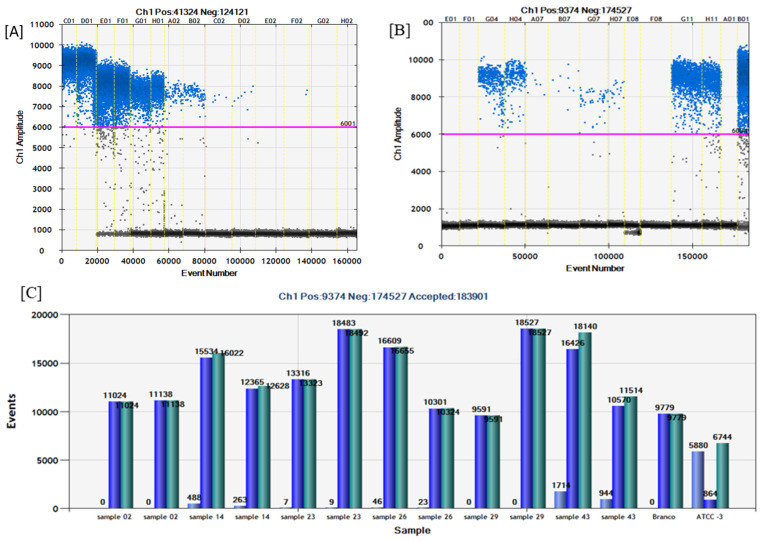
(**A**) Analytical sensitivity of ddPCR using serial dilutions of *A. baumanni* ATCC 19606 (reference sample). C01–D01: 10^2^. E01–F01: 10^3^. G01–H01: 10^4^. A02–B02: 10^5^. C02–D02: 10^6^. E02–F02: 10^7^. G02–H02: 10^8^. The blue droplets are considered positive, and the dark droplets are negative. Pink line: fluorescence amplitude cut-off. (**B**) ddPCR in clinical samples. A Fluorescence amplitude versus event number (total droplets). E01–F01: sample 02. G04–H04: sample 14. A07–B07: sample 23. G07–H07: sample 26. E08–F08: sample 29. G11–H11: sample 43. A01: no template control (NTC). B01: positive control *A. baumanni* ATCC 19606. The blue droplets are considered positive, and the dark droplets are negative. The pink line is the fluorescence amplitude cut-off (around 6000). (**C**) Number of total droplets (green), positive droplets (light blue), and negative droplets (dark blue) per sample.

**Figure 2 antibiotics-13-00878-f002:**
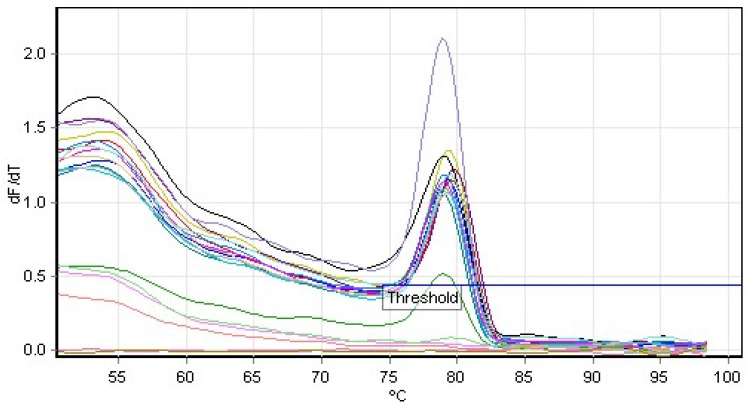
Melting curve analysis of real-time PCR (SYBR)’s analytical specificity. The values of positive control peaks (fluorescence over time higher than 2) and some bacteria (*S. aureus ATCC 25923, K. pneumoniae ATCC 27799, E. coli ATCC 25922, S. epidermidis 2ATCC 29213, P. aeruginosa ATCC 27853*, *and K. oxytoca 700324*) were similar (around 79 °C).

**Figure 3 antibiotics-13-00878-f003:**
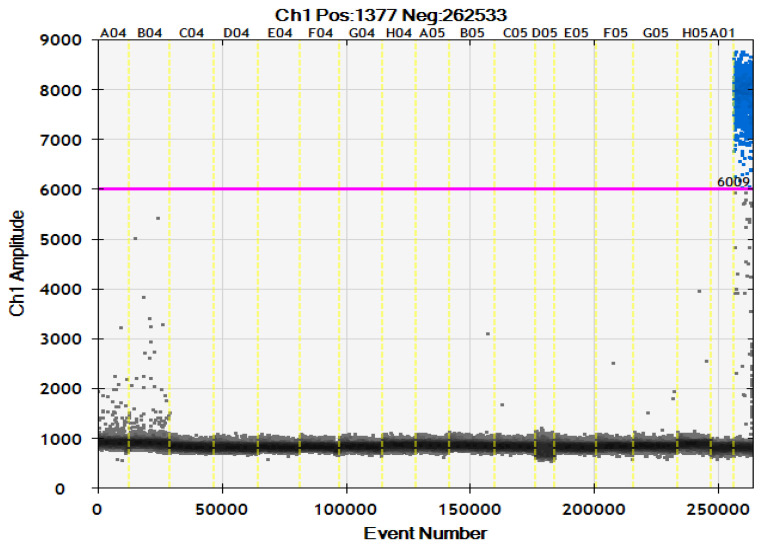
Analytical specificity of ddPCR using the panel of bacteria A04–B04: *S. aureus*, C04–D04: *P. aeruginosa*, E04–F04: *K. pneumoniae*, G04–H04: *E. coli*, A05–B05: *E. faecalis*, C05–D05: *K. oxytoca*, E05–F05: *E. faecium*, G05–H05: *S. epidermidis*. A01: no template control (NTC). Last column: positive control *A. baumanni* ATCC 19606. The blue droplets are considered positive, and the dark droplets are negative. Pink line: fluorescence amplitude cut-off.

**Table 1 antibiotics-13-00878-t001:** Amplification results of molecular assays by quantification cycle or absolute quantification belonging to the dilutions close to the limit of detection.

Sample	Dilutions	SYBR Cq	SYBR CV	TaqMan Cq	TaqMan CV	ddPCR Copies	ddPCR CV
ATCC 19606	10^5^	35.61	0.6	30.46	0.5	194	8.0
		35.17		30.59		228	
		36.06		30.93		-	
	10^6^	-	-	33.53	1.7	9.6	19.3
		-		33.72		14.2	
		-		34.93		-	
	10^7^	-	-	-		1.4	-
		-		-		-	
		-		-		-	
Clinical sample	10^5^	36.48	1.7	31.42	0.4	156	11.4
		37.08		31.62		124	
		38.28		31.80		-	
	10^6^	-	-	35.22	0.5	14	23.9
		-		35.22		8.6	
		-		35.68		-	
	10^7^	N.E	-	-	-	-	-
		N.A		-		-	
		N.A		-		-	

**Table 2 antibiotics-13-00878-t002:** Results of microbiological analysis and molecular assays (conventional PCR, SYBR^®^Green, TaqMan^®^, and ddPCR) in 44 mini-BAL samples.

No	Microbiological Analysis	Conventional PCR	SYBR^®^Green	TaqMan^®^	ddPCR	ddPCR
	Quantitative Culture (CFU/mL Count)	Electrophoresis’s Result	Cq Value	Cq Value	Number of + Droplets (Average)	Number of Copies/µL (Average)
1	-	-	28.80	-	-	-
2	-	-	32.21	-	-	-
3	-	-	29.90	-	-	-
4	-	-	29.10	-	-	-
5	40	-	28.95	-	-	-
6	-	-	30.70	-	-	-
7	-	-	29.20	-	-	-
8	-	-	29.37	-	-	-
9	350–400	+	10.10	25.14	1640	126.1
10	350–400	+	19.52	32.87	107	9.65
11	-	-	28.98	-	4	0.265
12	-	-	28.80	-	-	-
13	-	-	30.87	-	-	-
14	100	+	25.62	27.99	376	30.6
15	-	-	30.41	-	-	-
16	-	-	34.19	-	-	-
17	-	-	31.66	-	-	-
18	-	-	29.76	-	-	-
19	-	-	-	-	-	-
20	-	-	30.01	-	-	-
21	-	-	30.03	-	-	-
22	-	-	30.88	-	-	-
23	-	-	28.97	-	8	0.595
24	-	-	29.46	-	-	-
25	-	-	29.98	-	-	-
26	350–500	+	10.08	30.82	35	2.95
27	-	-	29.09	-	5	0.33
28	-	-	27.92	34.52	11	0.815
29	-	-	30.50	-	-	-
30	-	-	29.84	34.29	5	0.4
31	-	-	30.33	31.33	80	5.7
32	-	-	30.22	-	-	-
33	-	-	28.85	-	-	-
34	200	+	23.30	30.26	570	42.75
35	79	-	30.02	29.58	94	9.35
36	-	-	30.37	-	-	-
37	-	-	28.51	-	8	0.525
38	-	-	31.12	-	-	-
39	-	-	28.33	-	-	-
40	-	-	28.45	-	5	0.3
41	-	+	28.42	33.04	17	18.2
42	-	-	23.55	-	-	-
43	135	+	29.15	26.04	1329	108.7
44	-	+	21.27	-	11	0.895

**Table 3 antibiotics-13-00878-t003:** Number of samples detected and undetected in conventional PCR, Taqman^®^, and ddPCR relative to quantitative culture.

Molecular Assays		Quantitative Culture		Total	Kappa (%)/Error
		Detected	Non-Detected		
Conventional PCR	Detected	6	2	8	69.44/0.1183
	Not detected	2	34	46	
	Total	8	36	44	
Taqman	Detected	7	4	11	66.67/0.1125
	Not detected	1	32	33	
	Total	8	36	44	
ddPCR	Detected	7	10	17	41.55/0.1096
	Not detected	1	26	27	
	Total	8	36	44	

**Table 4 antibiotics-13-00878-t004:** Number of samples detected and undetected in ddPCR relative to conventional PCR and Taqman^®^.

Molecular Assays		ddPCR		Total	Kappa(%) Error
		Detected	Not Detected		
Conventional PCR	Detected	8	0	8	
	Not detected	9	27	36	52.17/
	Total	17	27	44	0.1036
Taqman^®^	Detected	11	0	11	
	Not detected	6	27	33	69.23/
	Total	17	27	44	0.0921

## Data Availability

Data are contained within the article.

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
