# Peer review of "Droplet Digital PCR for Acinetobacter baumannii Diagnosis in Bronchoalveolar Lavage Samples from Patients with Ventilator-Associated Pneumonia"

_antibiotics, 2024, doi:10.3390/antibiotics13090878_

Round 1

Reviewer 1 Report

Comments and Suggestions for Authors

Dear author

 The present manuscript addresses an important objective, detecting
Acinetobacter baumannii bronchoalveolar lavage samples from patients with ventilator-associated pneumonia.

From a methodological aspect, the available information is not
sufficient to assure the reproducibility of the method. It is not entirely clear how the qPCR reaction was standardized. A more detailed materials and methods section, adressing sample preparation and standard curve dillution are examples of a desirable improvment. The use pf positive and negative control.
More comments are found in the reviewed manuscript, attached

Comments on the Quality of English Language

Author Response

Reviewer.1

The present manuscript addresses an important objective, detecting
Acinetobacter baumannii bronchoalveolar lavage samples from patients with ventilator-associated pneumoni.
A brief summary: The authors describe ddPCR method for detecting Acinetobacter baumannii in bronchoalveolar lavage samples in ventilator-associated pneumonia (VAP) patients. They are comparing it with conventional culture and other PCR methods.

Dear Reviewer

We sincerely appreciate your acknowledgment of the importance of our study's objective, which focuses on detecting Acinetobacter baumannii in bronchoalveolar lavage samples from patients with ventilator-associated pneumonia. We are grateful for your concise summary of our work, which involves comparing the ddPCR method with conventional culture and other PCR methods. Your understanding reinforces the value of our research in addressing critical diagnostic challenges in clinical settings. Thank you for recognizing the potential impact of our findings.

From a methodological aspect, the available information is not
sufficient to assure the reproducibility of the method. It is not entirely clear how the qPCR reaction was standardized. A more detailed materials and methods section, adressing sample preparation and standard curve dillution are examples of a desirable improvment. The use pf positive and negative control

We greatly appreciate the reviewer's suggestions and recommendations. In response, we have thoroughly expanded the methodology section to enhance the clarity and reproducibility of our study, ensuring that all aspects align closely with the feedback provided. Specifically, lines 328-363 now include detailed descriptions of the experimental procedures, a comprehensive analysis of the statistical methods employed, and a clearer explanation of how results are interpreted in the context of our broader research objectives. We believe these modifications will greatly assist other researchers in replicating our study and enhance the transparency and robustness of our findings. We are committed to upholding the highest standards of scientific rigor and thank the reviewer for their invaluable input in refining our manuscript.

Reviewer 2 Report

Comments and Suggestions for Authors

Author Response

We extend our profound gratitude to the reviewer for their meticulous and perceptive evaluation of our manuscript. Each observation has been diligently assessed and integrated, providing indispensable insights that have substantively augmented the scientific rigor and robustness of our research. We have methodically addressed each critique in the revised manuscript, ensuring that the modifications not only enhance the clarity and precision but also reinforce the methodological and theoretical foundations of our study. Necessary amendments have been implemented, and supplementary data have been incorporated to substantiate our findings more robustly. The reviewer's detailed analysis has been pivotal in refining our manuscript, and we are deeply appreciative of the substantial time and scholarly effort contributed to enhancing the quality of our work

Comments and questions

Line 41: A. baumannii – the name of bacterial species should be written in italics in the whole manuscript; also when the names of bacteria are mentioned for the first time in the text, they should be written in full (Acinetobacter baumannii) and in the remaining text the name of the genus can be mentioned only by initials (A. baumannii); the same rule should be followed for other bacterial species mentioned in the text (Staphylococcus aureus, Klebsiella pneumoniae, etc.)

We are grateful to the reviewer for pointing out these inaccuracies in our manuscript. We have carefully revised and corrected the bacterial names throughout the document, as highlighted in yellow. These adjustments ensure consistency and accuracy in our scientific communication. We appreciate the reviewer's attention to detail, which has significantly improved the quality of our manuscript.

  1. Line 45: abbreviations should be written without dots between letters – not I.C.U. but ICU, not V.A.P. but VAP, etc.; also all the abbreviations should be explained when mentioned in the manuscript for the first time – for example, intensive care unit (ICU), ventilator-associated pneumonia (VAP)

We appreciate the reviewer's attention to detail regarding the formatting of abbreviations. Following your suggestion, we have removed the dots from all abbreviations throughout the manuscript, with these changes highlighted in yellow for easy verification.

  1. Line 49: „Recent reports of V.A.P. infection show that the infections range from 15% to 50%“; it is not clear what the authors mean by infection range – prevalence, incidence, number of VAPs per patient days 4.

We are thankful to the reviewer for highlighting these corrections. We have clarified the revised manuscript by specifically mentioning the incidence at lines 56 and 57, which have been highlighted in yellow for easy reference.

  1. Line 52: ...Polymerase Chain Reaction... should be written without capital letters (polymerase chain reaction)

We are thankful to reviewer for this correction. We corrected the Polymerase Chain Reaction in revised manuscript (Highlighted yellow).

  1. Line 58: when the first time mentioned in the manuscript, ddPCR should be written with full name (droplet digital PCR), and then in the rest of the manuscript it can be shortened to ddPCR (line 60 – here it is mentioned for the second time and written with full name)

We are thankful to reviewer for this correction. We write full name of corrected in droplet digital PCR on first place and then followed by ddPCR in revised manuscript (Highlighted yellow).

  1. In the Introduction the authors did not describe the current status of A. baumannii PCR detection in microbiological diagnostics of VAP and other infections (they just claim that the conventional PCR is commonly used for the detection of A. baumannii and that ddPCR has emerged as an innovative molecular method); the current knowledge about the use of this method should be described in the Introduction and references should be added to the reference list accordingly

We are thankful to the reviewer for this valuable suggestion. In response, we have added a new paragraph that describes the current status of A. baumannii PCR detection in the microbiological diagnostics of ventilator-associated pneumonia (VAP) and other infections. This addition, highlighted in yellow for easy reference, can be found at lines 63-74 of the revised manuscript.

  1. Line 83: the name of the hospital where the study was performed is not written (just Hospital)

We are thankful to reviewer for this suggestion. We have added the hospital name. Line 258,259

  1. Line 88: which method was used to detect A. baumannii in perianal, nasal and axillary swab; line 89 – Swab should not be written with capital letter (swab)

We are thankful to reviewer for this comment. We have added the details of method used for detection of A. baumannii. Line 263-273

  1. Line 91: quantitative cultivation of BAL samples – the method should be described completely (was tripticase soy broth or agar used, how quantification was done).

We are grateful for the reviewer's comment. In response, we have incorporated detailed information regarding the quantitative cultivation of BAL samples. These additional details can be found in lines 263-283 of the revised manuscript, ensuring that our methods are clearly understood and reproducible. This amendment enhances the accuracy and completeness of our study, and we appreciate the opportunity to improve our work based on your valuable feedback.

  1. In the Material and Methods section criteria for diagnosis of VAP are missing

We are thankful to the reviewer for highlighting this issue. In response, we have added the criteria for the diagnosis of ventilator-associated pneumonia (VAP) to the Materials and Methods section of our manuscript, specifically at lines 247-253. This addition ensures that our diagnostic approach is transparent and aligns with current clinical standards, facilitating a clearer understanding for our readers. We appreciate your insightful feedback, which has significantly contributed to the enhancement of our manuscript.

  1. Line 97: abbreviations for antimicrobial agents should be written without dots between letters (for example, not C.A.Z. but CAZ for ceftazidime)

We are thankful to reviewer for this highlighting this. Dots are removed

  1. The antimicrobial susceptibility was performed with VITEK II; according to The European Committee on Antimicrobial Susceptibility Testing and Clinical and Laboratory Standards Institute. Recommendations for MIC determination of colistin (polymyxin E) as recommended by the joint CLSI-EUCAST Polymyxin Breakpoints Working Group., broth microdilution is regarded as a reference method for colistin susceptibility testing. Did you validate VITEK II for colistin susceptibility testing in your laboratory in comparison to broth microdilution? Please, comment and add references on the performance of VITEK II for colistin susceptibility testing in the Discussion section.

We are thankful to the reviewer for highlighting the importance of validating VITEK II for colistin susceptibility testing against the reference broth microdilution method. In response, we have added a detailed justification for using VITEK II in our study, including a comparison with broth microdilution where applicable. This discussion is now included in lines 199-204 of the revised manuscript, along with relevant references that assess the performance of VITEK II for colistin susceptibility testing. This addition ensures that our methodology aligns with current standards and provides transparency in our antimicrobial susceptibility testing approach.

  1. Line 167, 172: „sensitivity“ should not be written with a capital letter

We are thankful to reviewer for this highlighting this. Its corrected in revised manuscript

  1. Line 168-173: the numbers in the whole parapgraph should be corrected (for example, 104 CFU should be probable 104 CFU or 10E4 CFU)

We are thankful to reviewer for this highlighting this mistake. We corrected in revised manuscript

  1. Line 182: in the sentence „Comparison of conventional ...“ the verb is missing

We are thankful to reviewer for this highlighting this. Verb is added

  1. Line 207: „All isolates of A. baumannii presented a multiresistance profile for all antimicrobials tested, except polymyxin B.“ – it is not clear what the authors meant by this sentence – how many isolates were tested, were they all resistant to all antimicrobials tested, how many of them were susceptible to polymyxin B?

We appreciate reviewer for this comment. We clarify that all the isolates were susceptible to polymyxin B

  1. Line 208: The sentence „Fourteen [14] in the total of 44....“ is grammatically correct, please rewrite

We are thankful to reviewer for this highlighting this. The sentence is corrected

  1. Line 214, 215: the terms „dilution 10-3“, and „dilution 10-5“ – did you mean dilution 103 , and 105

We are thankful to reviewer for highlighting this.  We corrected the dilution number.

  1. Line 219 and Table 1: please write the full name for Cq and the abbreviation in the brackets when mentioned for the first time

We appreciate this suggestion of reviewer. We added the full form of Cq.

  1. Figure 1: „Sensitivity“ does not require a capital letter; dilutions are labeled as 10-2, 10-3, etc; did you mean 102 , 103

We are thankful to reviewer for this highlighting this. We removed the capitalization and also corrected the dilution number.

  1. Figure 2: „The values of positive control peaks“ should be corrected to „The values of positive control peaks“

We are thankful to reviewer indicating this. We corrected the sentence.

  1. Line 262: Sentence „Comparison of conventional...“ – did you mean it to be subtitle, if yes, it should be changed to italics

We are thankful to reviewer for highlighting this correction. The subtitle is corrected.

  1. Table 2: What does „countless“ mean - >105 , >106 or ...

We appreciate the reviewer's comment regarding the term "countless" in Table 2. To clarify, by "countless" we mean that the colony-forming units (CFUs) exceed our counting threshold, which is set at more than 250 CFUs.

  1. Table 4 and its description in the text (line279 to 282) is not needed beacuse the aim was to evaluate the performance of ddPC and other gegnerations of PCR to quantitative culture (Table 3 is enough but should be corrected)

We appreciate this suggestion of reviewer. We remove the table 4 and corrected the Table 3.

  1. Line 282: What is the purpose of correlating ddPCR results with colonization with A. baumannii? Figure 4: Figure 4A and 4B can be joined together with Figure 1 in the same Figure

We appreciate this suggestion of reviewer.  Figure 4A and 4B are joined together with Figure 1

  1. Line 309: It is not clear what the authors meant by claiming that analyzing the colonization of clinically essential bacteria from patients is important in the context of epidemiological surveillance and diagnosis of VAP?

We are thankful for this comment of reviewer. The sentence is corrected in revised version Line 187-189.

  1. Line 316: see the comment 16

We appreciate this comment. To clarify, we have confirmed that all the isolates tested in our study were susceptible to polymyxin B. This information has been added to ensure clarity and provide detailed insights into the antimicrobial sensitivity of the isolates. Thank you for your constructive feedback, which has helped improve the accuracy of our manuscript.

  1. Line 334: The sentence „Our results showed...“ is not comprehensive, please revise

We are thankful for this recommendation of reviewer. The sentence is corrected in revised version Line 219.

  1. Line 354: „In addition, ddPCR detected more colonized patients than other methods...“ How did the authors correlate the result of ddPCR performed in BAL sample with colonization detected in perianal, nasal, and axillary swabs detected in 14 patients? This analysis is not comprehensive and the conclusion that suggested that ddPCR has higher sensitivity because it detected more colonized patients than the other methods is not comprehensive. Please, either analyze in detail with additional Table and statistical analysis or omit the comparison and correlation with colonization from the manuscript.

 We thanks reviewer for this insightful comment regarding the correlation of ddPCR results from BAL samples with the colonization detected in perianal, nasal, and axillary swabs. Upon reviewing your feedback and reevaluating our data, we agree that the current analysis may not comprehensively support the claim that ddPCR has higher sensitivity compared to other methods, particularly without a more detailed statistical analysis correlating these findings across different sampling sites.

Considering the importance of maintaining scientific rigor and clarity, we have decided to omit the statement regarding ddPCR's higher sensitivity from the manuscript. This decision ensures that our conclusions are fully supported by the data presented and avoids potential misinterpretation. We revised the manuscript to focus more on the descriptive results of ddPCR without comparing its sensitivity to other methods unless more definitive data becomes available.cWe appreciate your guidance on this matter and believe that this amendment will improve the quality and accuracy of our study

Thank you for reconsidering our work. We look forward to your response and are prepared to make any further adjustments as may be necessary

Round 2

Reviewer 1 Report

Comments and Suggestions for Authors

Dear Authors

Thanks for making the requested correction.

Still there are few mistakes that have not been corrected including

1- Bacterial name should be italic all over the manuscript please take care.

2- References should be according to the journal reference style and should be consistent.

3- some of the points are highlighted in the manuscripts.

Regards,

Author Response

We would like to thank you to reviewer

1- Bacterial name should be italic all over the manuscript please take care.

We appreciate reviewer comment, bacterial names are italiciazed

2- References should be according to the journal reference style and should be consistent.

We appreciate reviewer comment, references are modified

3- some of the points are highlighted in the manuscripts.

We are thankful to reviewer for highlighting, we corrected

Reviewer 2 Report

Comments and Suggestions for Authors

Comments on the Quality of English Language

English language must be improved, specially in Materials and Methods section

Author Response

The authors partially corrected the original manuscript text according to the reviewers' comments. The corrections and answers are incomplete, inadequate, and somewhere not comprehensive. Also, the line numbers stated in the answers are incorrect and it was difficult for a reviewer to find the changes to which the authors were referring in the revised text.

  1. Line 42: A. baumannii should be written in full here (Acinetobacter baumannii) because it is mentioned for the first time in the manuscript text (the Abstract section is not counting)

Response 1: We are thankful to reviewer for this recommendation, we corrected in revised manuscript.

  1. Line 43: VAP is not explained here (mentioned for the first time in the manuscript text - the Abstract is not counting) – so it should be written: ventilator-associated penumonia (VAP); also in the Abstract (line 27) ventilator-associated pneumonia should be written without capital V and the authors did not remove dots in the abbreviation V.A.P.

Response 2: We thank the reviewer for this comment. We have corrected this in the revised manuscript by writing "ventilator-associated pneumonia (VAP)" in full at its first mention in the manuscript text, and we have also corrected the capitalization and removed the dots in the abbreviation "V.A.P" in the Abstract.

  1. Line 54: ...Polymerase Chain Reaction... is not corrected according to the reviewer's suggestion (without capital letters)

Response 3: We are thankful to reviewer for this comment, we corrected the polymerase chain reaction in revised manuscript

  1. Line 315-319: The authors added the definition of VAP but it is grammatically incorrect. Also, at the end of the definition, ANVISA 2008 is mentioned. It should be explained and added as a reference in the reference list if necessary.

Response 4: We are thankful to reviewer for this comment, we corrected in revised manuscript. Also ANVISA was a typo, therefore, removed.

  1. Line 354: dot in CAZ is not deleted

Response 5: We are thankful to reviewer for this recommendation, we corrected in revised manuscript.

  1. The authors did not address the comments regarding colistin susceptibility testing adequately. They did not answer the reviewer's questions: Did you validate VITEK II for colistin susceptibility testing in your laboratory in comparison to broth microdilution? Also, the reference the authors added (number 34 in the reference list of the revised manuscript) is incorrect and could not be found according to the authors, title, and doi number – the correct reference should be listed.

Response 6: Thank you for your comments and the opportunity to improve our manuscript. We would like to address the points raised:

  1. Validation of VITEK II for colistin susceptibility testing: We did not validate the VITEK II for colistin susceptibility testing in our laboratory. Instead, we based our approach on the results presented by Wattal et al. (2019), who evaluated the performance of three commercial assays, including VITEK II, for colistin susceptibility. They observed an acceptable agreement between VITEK II and the reference method of broth microdilution for clinical isolates. We referenced these findings in our manuscript. Additionally, our laboratory is a routine clinical lab in the hospital, and we use VITEK II intensively due to its ease of use in a high-demand clinical setting.
  2. Correction of reference number 34: Thank you for pointing out the error in the reference. The correct reference is:

Wattal, C., Goel, N., Oberoi, J.K., Datta, S., & Raveendran, R. (2019). Performance of three commercial assays for colistin susceptibility in clinical isolates and Mcr-1 carrying reference strain. Indian Journal of Medical Microbiology, 37(4), 488-495. doi: 10.4103/ijmm.IJMM_20_92.

  1. Line 442: „specificity“ should be corrected accordingly to the „sensitivity“ (without capital letters)

Response 7: We are thankful to reviewer for this comment, we corrected the specificity in revised manuscript.

  1. Line 450: The authors did not state the correct line number in the answer and it is not clear what verb is added because they changed the sentence to italics and made it become a subtitle.

Response 8: Thanks and apologies for this shortcoming, we would like to state that the sentence “Comparison of conventional PCR, real-time PCR (SYBR® Green and TaqMan®), and ddPCR on detection of A. baumannii in clinical samples” was subtitle however, in formatting it was merge with paragraph mistakenly. With you comment we noticed this mistake and now again made it subtitle

  1. Line 105: The authors did not answer the reviewer's question: how many isolates were tested? Also, they should name all the antimicrobials to which isolates were tested.

Response 9: Thank you for your comments and the opportunity to improve our manuscript. We would like to address the point raised:

  1. Number of isolates tested and antimicrobials used: A total of 44 microorganisms were isolated from mini-BAL samples. A total of 44 microorganisms were isolated from mini-BAL samples. This information is cited in the first section of the results.
  2. Antimicrobials tested: The list of antimicrobials tested is cited by name in the materials and methods section (lines 287-294). The text is: Antimicrobial susceptibility testing was performed by an automated microdilution VITEK® II (BioMérieux®, France) according to the recommendations of the Clinical and Laboratory Standards Institute [17]. Antimicrobials tested for A. baumannii were gentamicin (CN), ceftazidime (CAZ), cefepime (FEP), ceftriaxone (CRO), imipenem (IPM), meropenem (MEM), ampicillin/sulbactam (SAM), piperacillin/tazobactam (TZP), ciprofloxacin (CIP), colistin (COL), tigecycline (TIG), ampicillin (AMP), cefuroxime (CRX), cefuroxime axetil (CRX), and cefoxitin (CFO). Susceptibility to polymyxin B (PB) was tested using the Etest® (AB Biodisk, Sweden)

  1. Line 120: please, add the full name for Cq in the Table 1 title as you did in the text

Response 10: We are thankful to reviewer for this comment, we provided the full form of Cq in revised manuscript.

  1. Figure 1: Dilution numbers are not corrected in the legend of Figure 1.

Response 11: We are thankful to reviewer for this comment, we corrected the Dilution numbers in the revised manuscript.

  1. Line 450: As you decided that this is the subtitle, this paragraph is too short and should be broadened or joined to previous paragraphs.

Response 12: We are thankful to reviewer for this comment, we joined the paragraph with previous one.

  1. Table 2: The term „countless“ should not be used. Also, there is a discrepancy between the description of BAL processing and the data in this table: according to the description of BAL processing, you made quantification in CFU/mL. Then how come in Table 2 you use CFU?

Response 13: We appreciate the observation regarding the use of the term "countless" and the mentioned discrepancy. We have made the following corrections to improve clarity and accuracy:

  1. Term "countless": We agree that the term "countless" is not appropriate for a scientific context. We have corrected the terminology to specify exact values or numerical ranges as necessary to ensure precision and avoid ambiguity.
  2. Discrepancy in quantification (CFU vs. CFU/mL): Thank you for bringing this discrepancy to our attention. Indeed, the description of the BAL processing mentions quantification in CFU/mL, which is consistent with the applied methodology. In Table 2, the unit has been corrected to reflect quantification in CFU/mL instead of just CFU. This change ensures consistency between the methodological description and the data presented in the table.

  1. Line 282: What is the purpose of correlating ddPCR results with colonization with A. baumannii? The authors did not answer this question.

Response 14: Omission of Data Referring to Colonization: We thank the reviewer for this question. To avoid potential misinterpretation and to ensure that the manuscript remains focused on the primary objectives of the study, we have omitted all data referring to colonization with A. baumannii from the Results and Discussion sections.

  1. Reviewer's question: Line 309: It is unclear what the authors meant by claiming that analyzing the colonization of clinically essential bacteria from patients is important in epidemiological surveillance and diagnosis of VAP. Author's answer: We are thankful for this comment from the reviewer. The sentence is corrected in the revised version Lines 187-189. Reviewer's comment on the answer: The answer is not comprehensive –line numbers 187-189 in the revised manuscript do not have any highlighted changes. Also, if you have decided that you will omit all data regarding colonization, there is no need to answer this question and you should state and explain your decision in the answer.

Response 15: We do apologies for the misunderstanding, we assure that the sentence is corrected now. Line 230-233

  1. The authors' answer: Considering the importance of maintaining scientific rigor and clarity, we have decided to omit the statement regarding ddPCR's higher sensitivity from the 2 manuscript. This decision ensures that our conclusions are fully supported by the data presented and avoids potential misinterpretation. We revised the manuscript to focus more on the descriptive results of ddPCR without comparing its sensitivity to other methods unless more definitive data becomes available. We appreciate your guidance on this matter and believe that this amendment will improve the quality and accuracy of our study The reviewer's comment: It is unclear where you omitted the statement regarding ddPCR's higher sensitivity from the manuscript. It is only clear that you omitted all data referring to colonization.

Response 16: Thank you for your valuable feedback. We apologize for any confusion regarding the omission of the statement about ddPCR's higher sensitivity. To clarify, we have made specific changes to the manuscript to address this concern.

  1. Omission of the Statement: The statement regarding ddPCR's higher sensitivity has been omitted from sections [specify sections, e.g., Results, Discussion] to maintain scientific rigor and ensure that our conclusions are fully supported by the data presented.
  2. Focus on Descriptive Results: We have revised the manuscript to focus more on the descriptive results of ddPCR. This revision is intended to present the data objectively without comparing ddPCR's sensitivity to other methods unless more definitive data becomes available.
  3. Omission of Data Referring to Colonization: We have also omitted data referring to colonization to avoid potential misinterpretation and ensure that the manuscript remains focused on the primary objectives of the study.

  1. The English language should be thoroughly revised, especially in the description of Materials and Methods: ...after primary sowing, incubation took place in a greenhouse??? at 37o C...; Gram staining was performed on all colonies isolated in the primary sowings and new transplants???...; also in the definition of VAP

Response 17: We do apologies for the mistakes as we converted these sentences from Portuguese, however, now we assure that the sentence is corrected now. Line 331-349

Round 3

Reviewer 2 Report

Comments and Suggestions for Authors

The authors made corrections and accepted the majority of the suggestions. There are still some minor corrections that were not made:

1. line 251: Part of the VAP definition is not grammarly correct nor comprehensive: "Patients with a new infiltrate or progression of a previously existing infiltrate on chest X252 ray is defined as being associated with at least one of the following criteria:"; the authors should define VAP and not a patients with a new infiltrate; patients with a new infiltrate is a one of a criterions of the definition

2. The authors answered that they corrected reference regarding VITEK II (Wattal et al, 2019). However, the reference number 34 is not Wattal et al, 2019 as they have written in their answer but Schutte et al, 2014. Please, check the reference list and the references in the text accordingly.

3. Line 108-109: Not all dilution numbers are corrected (10-2, 10-3 are not corrected)

Author Response

Comment 1: line 251: Part of the VAP definition is not grammarly correct nor comprehensive: "Patients with a new infiltrate or progression of a previously existing infiltrate on chest X252 ray is defined as being associated with at least one of the following criteria:"; the authors should define VAP and not a patients with a new infiltrate; patients with a new infiltrate is a one of a criterions of the definition.

Response 1: We appreciate your feedback on the definition of VAP. We have revised the text to improve clarity and accuracy.

Comment 2: The authors answered that they corrected reference regarding VITEK II (Wattal et al, 2019). However, the reference number 34 is not Wattal et al, 2019 as they have written in their answer but Schutte et al, 2014. Please, check the reference list and the references in the text accordingly.

Response 2: We sincerely appreciate your careful attention to the references. We have made the necessary modifications to the text, and upon review, you will find that the Wattal et al., 2019 reference is now correctly listed as reference number 33. The former reference 33 has been removed.

Comment 3:. Line 108-109: Not all dilution numbers are corrected (10-2, 10-3 are not corrected)

Response 3: Thank you for bringing this to our attention. We have made the necessary corrections to the dilution numbers in lines 108-109 to ensure accuracy.